# The Greenhouse Phyllosphere Microbiome and Associations with Introduced Bumblebees and Predatory Mites

Marie Legein,[a] Wenke Smets,[a] Karen Wuyts,[a] Lien Bosmans,[b] Roeland Samson,[a] Sarah Lebeer[a]

[a]Environmental Ecology and Applied Microbiology (ENdEMIC), Department of Bioscience Engineering, University of Antwerp, Antwerp, Belgium
[b]Research Centre Hoogstraten vzw, Meerle, Belgium

**ABSTRACT** Greenhouses are highly productive environments in which conditions are regulated to optimize plant growth. The enclosed character of greenhouses usually results in reduced microbial diversity, while it is known that a diverse microbiome is important for plant health. Therefore, we explored the phyllosphere microbiome of tomatoes and strawberries grown in greenhouses. We observed that the microbiome of both crops was low in diversity and abundance and varied considerably over time and space. Interestingly, the core taxa of tomatoes were *Snodgrasella* and *Gilliamella*, genera typically associated with bumblebees. The same amplicon sequence variants (ASVs) were found on reared bumblebees, indicating that the bumblebees, present in the sampled greenhouses to pollinate flowers, had introduced and dispersed these bacteria in the greenhouses. Overall, we found that 80% of plants contained bumblebee-associated taxa, and on these plants, bumblebee-associated reads accounted for up to a quarter of the reads on tomatoes and a tenth of the reads on strawberries. Furthermore, predatory mites had been introduced for the control of spider mites. Their microbiome was composed of a diverse set of bacteria, which varied between batches ordered at different times. Still, identical ASVs were found on mites and crops, and these belonged to the genera *Sphingomonas*, *Staphylococcus*, *Methylobacterium*, and *Pseudomonas*. These new insights should now be further explored and utilized to diversify ecosystems that are characterized by low diversity and abundancy of microbes.

**IMPORTANCE** Greenhouses, though highly effective agricultural environments, are characterized by reduced sources of bacterial diversity and means of dispersal compared to more natural settings. As it is known that plant health and productivity are affected by associated bacteria, improving our knowledge on the bacterial communities on greenhouse crops is key to further innovate in horticulture. Our findings show that tomato and strawberry crops cultivated in greenhouses harbor poor and variable bacterial communities. Furthermore, commonly implemented biological solutions (i.e., those based on living organisms such as bumblebees and predatory mites) are important sources and means of dispersal of bacteria in greenhouses. This study shows that there is great potential in using these biological solutions to enrich the greenhouse microbiome by introducing and dispersing microbes which have beneficial effects on crop production and protection, provided that the dispersed microbes have a beneficial function.

**KEYWORDS** phyllosphere, microbiome, predatory mites, bumblebees

Address correspondence to Sarah Lebeer, sarah.lebeer@uantwerpen.be.

The authors declare no conflict of interest.

Greenhouses are environments in which conditions such as temperature, humidity, and light intensity are controlled to optimize plant growth. Additionally, biological solutions (i.e., solutions based on living organisms) are commonly implemented, such as bumblebees (e.g., *Bombus terrestris*) used for pollination and predatory mites (e.g., *Phytoseiulus persimilis*) for the control of spider mites. Greenhouse technology is essential for food production, as it can result in substantially increased yields and a longer growing season (1). However, using controlled environments to grow crops generally

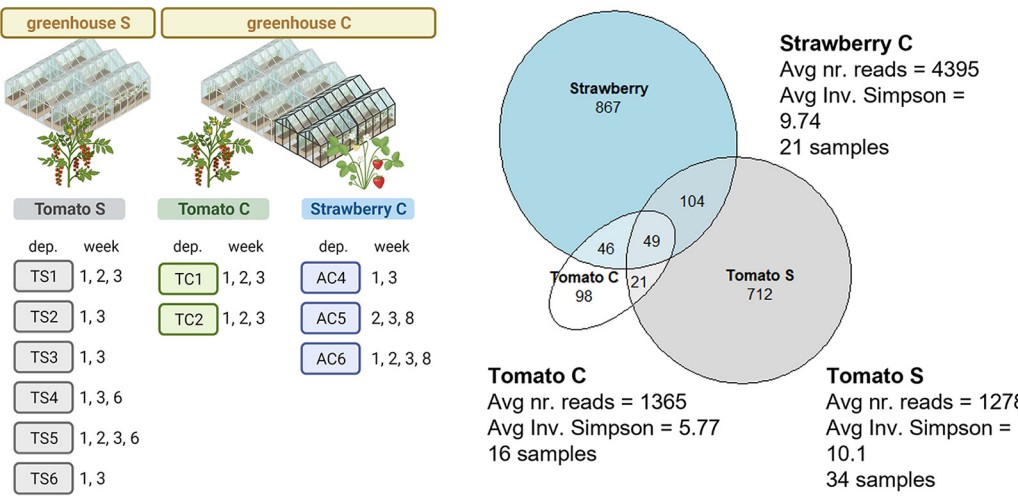

**FIG 1** (A) Sampling setup. Samples from tomato and strawberry plants were taken from two greenhouses, subdivided into departments. One to three samples were taken per department and per time point (weeks 1, 2, 3, 6, and 8). (B) Venn diagram showing the number of different amplicon sequence variants (ASVs) found in three groups: tomato plants from greenhouses C and S as well as strawberry plants from greenhouse C. The average number of reads, average inverse Simpson index, and number of samples are also indicated in the figure. ASVs were considered present in a sample if more than one read was present.

results in less diverse phyllosphere microbial communities (2–4). It has been suggested that this could be explained by a reduction in bacterial sources, such as soil, insects, and surrounding vegetation (4), as well as lower dispersal rates, as crops are not exposed to dispersing forces, such as wind or rain (3). Such a reduction in microbial diversity on greenhouse crops is of concern, as research increasingly indicates that bacterial communities associated with plants are important for plant health (5–7).

This study aimed to further explore the taxonomic composition of bacterial communities on greenhouse crops and their dynamics over time and space. More specifically, we explored the impact of introduced organisms, such as bumblebees and predatory mites, on the phyllosphere microbiome, as these arthropods could be a source and a means of dispersal of microbes in commercially operating greenhouses.

For this purpose, the phyllosphere bacterial communities on tomato (*Lycopersicum solanum*) and strawberry (*Fragaria x ananassa*) crops were sampled from two commercially operating greenhouses that implemented integrated pest management (IPM) in Flanders over the course of 8 weeks. The samples were analyzed using 16S rRNA gene amplicon sequencing. In these greenhouses, reared bumblebees and predatory mites had been introduced for pollination and the control of spider mites. Additionally, the contact microbiomes of commercial-reared bumblebees (*Bombus terrestris*) and predatory mites (*Phytoseiulus persimilis*) originating from the same breeding facilities as the arthropods used in the greenhouses were analyzed. The sampled bumblebees and mites had never come into contact with the sampled crops; however, the crops did come into contact with arthropods from the same breeding facilities.

## RESULTS

**Phyllosphere microbiome of greenhouse crops.** A total of 97 phyllosphere samples from strawberry (*n* = 28) and tomato plants (*n* = 69) were collected at five time points in March and April 2019. Tomato samples were collected from two commercially operating research greenhouses in Flanders (Research Centre for Vegetable Production vzw [S] and Research Centre Hoogstraten [C]), and strawberry samples were collected only from greenhouse (C), resulting in three sample groups (Fig. 1A). The two greenhouses were subdivided by departments and separated by glass walls. One to three samples were taken per department and per time point (weeks 1, 2, 3, 6, and 8), and these samples were sequenced using 16S rRNA amplicon sequencing. Nonbacterial

reads, of which the majority were mitochondrial and chloroplast sequences, were discarded. Samples were sequenced in three sequencing runs, and peptide nucleic acid (PNA) clamps were added in later runs to inhibit the amplification of mitochondrial and plastid DNA from the host (see Materials and Methods). On average, the percentage of nonbacterial reads in the first run, where PNA clamps were not used, was 96.6%. This decreased to 39.6% when both plastid and mitochondrial PNA clamps were used (see Materials and Methods). However, the use of PNA clamps also resulted in a low amplicon yield after PCR, and final bacterial read counts remained low (see Materials and Methods). After sequence processing and quality control, our data set consisted of 71 phyllosphere samples with a total of 157,589 bacterial reads. For 12 tomato and 2 strawberry samples, the leaf washes were plated out on nonselective Reasoner's 2A (R2A) agar medium. On average, strawberry leaves contained 650 $\pm$ 109 CFU/g and tomato leaves contained 975 $\pm$ 557 CFU/g.

Next, we explored the community composition on these plants. We first compared the number of amplicon sequence variants (ASVs) between the greenhouses. A higher number of different ASVs were found in the tomato samples collected from greenhouse S (886 ASVs, $n = 34$) than in the samples from greenhouse C (214 ASVs, $n = 16$). However, the inverse Simpson diversity index was not significantly different between tomato samples from the two greenhouses (5.77 [C] and 10.1 [S], $P = 0.516$) (Fig. 1B). For the three sample groups studied (i.e. strawberries in greenhouse S and tomatoes in greenhouses C and S), 49 ASVs were shared.

Next, we looked at the beta diversity between these samples (Fig. 2A and B) and the taxonomic composition of the samples (Fig. 2D and E). The tomato samples from department 2 from greenhouse C (TC2) clustered together in the PCoA plot, indicating that their phyllosphere communities were distinctive from those of other departments and that their communities remained similar throughout time. The barplot shows that the genus *Acinetobacter* was abundant in this department in all time points, while it did not occur in high abundances in other samples (Fig. 2C). A second cluster can be found on the left side of the PCoA plot. This cluster is comprised of a group of eight tomato samples from various departments and various time points. A closer look at these samples reveals that the genera *Snodgrassella* and *Gilliamella* were abundant there. Next, we observed that the genus *Portiera* was abundant in the tomato phyllosphere in departments TS1 and TS2 at various time points. In the strawberry samples (Fig. 2B), the genus *Erwinia* was present in high abundances only in one department (AC5) and at one time point (week 8).

A permutational multivariate analysis of variance (PERMANOVA) (Table 1) showed that department and week both had significant effects on the phyllosphere community composition. Permutational analyses for the homogeneity of multivariate dispersions (PERMDISP2) analyses were done to test whether these factors were not merely significant because they affected the size of the community variation rather than the average composition. PERMDISP2 confirmed the homogeneity of dispersion among groups defined by department and week for both the tomato and the strawberry samples (all $P$-values greater than 0.05) (Table 1). However, there was a difference in variance for the factor 'greenhouse' for the tomato samples ($P = 0.014$), meaning that the effect of the greenhouse factor was not necessarily due to a difference in average composition.

To further explore which taxa were consistent across departments and contributed to community composition, core ASVs were identified. Taxa were ranked based on their occupancy per department and their contribution to beta diversity (based on [8]). For the tomatoes, two ASVs from the genera *Snodgrasella* and *Gilliamella* were identified as core ASVs. For the strawberries, the core ASVs belonged to the genera *Clostridium*, *Pseudomonas*, *Terrisporobacter*, *Flavobacterium*, and *Janthinobacterium*. Their locations on the occupancy-abundancy distributions are shown in Fig. 3.

**Contact microbiome of commercially-reared bumblebees and predatory mites.** Both bumblebees (for pollination) and predatory mites (for the control of spider mites)

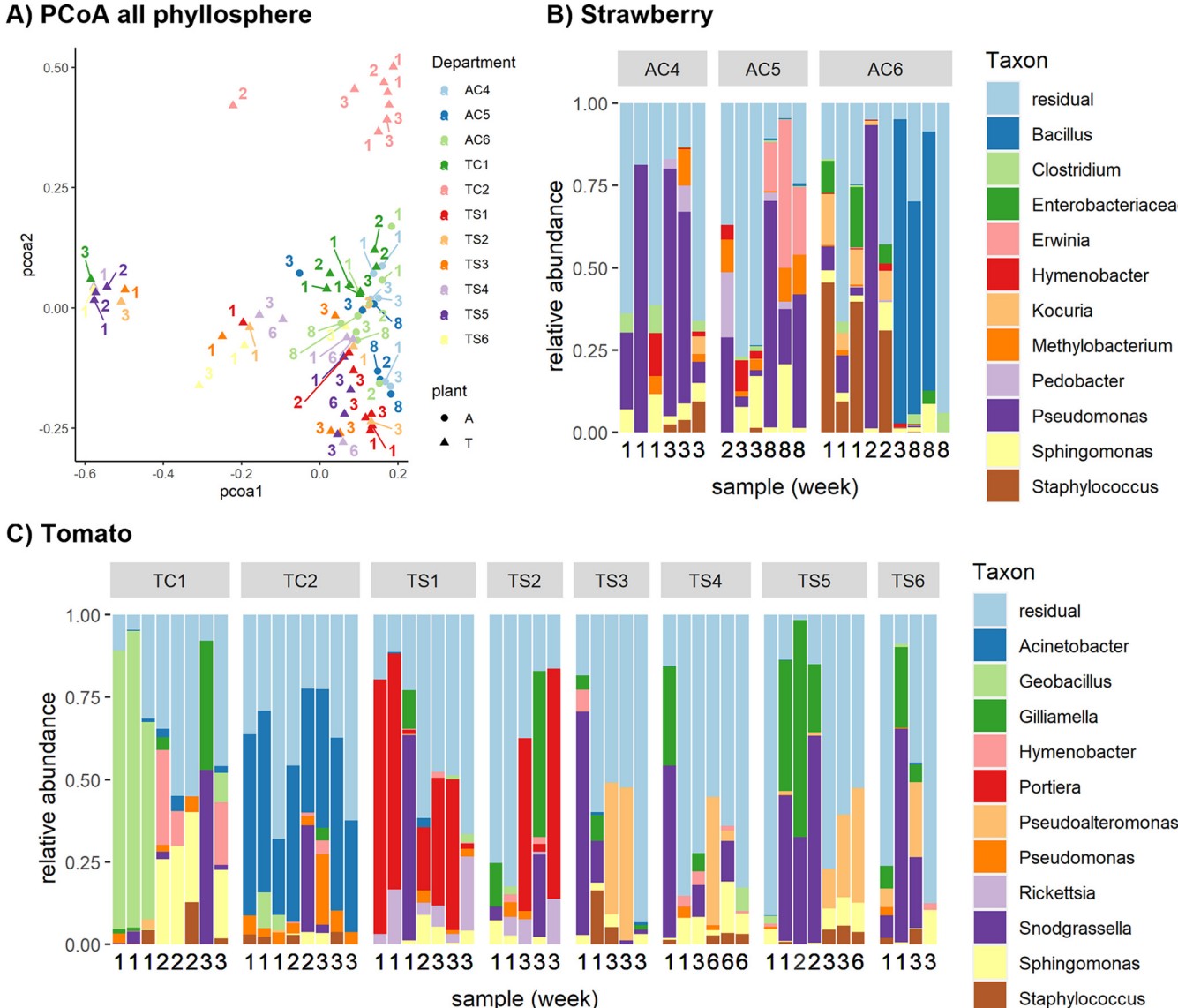

**FIG 2** Phyllosphere microbiome of strawberry and tomato plants. (A) PCoA plot visualizing the variation in the bacterial community composition of phyllosphere samples in a two-dimensional space based on the Bray-Curtis dissimilarity. Strawberry samples are indicated by circles and department codes starting with "A," while tomato samples are indicated by triangles and department codes starts with "T". Samples are colored by department, and sampling week is indicated by the number next to each symbol. (B and C) Barplots showing the 11 most abundant genera for each phyllosphere sample (strawberry in [B] and tomato in [C]). Samples are divided by department, and the sampling week is indicated by a number underneath each bar.

were present in the greenhouses where the phyllosphere samples were taken, as is common in greenhouses with IPM in Flanders. As a reference, bumblebees and predatory mites were also ordered from the same breeding facilities as the arthropods used in the greenhouses. The sampled bumblebees and mites had never come into contact with the greenhouse crops; however, the sampled plants did come into contact with bumblebees and mites that originated from the same breeding facilities. Their contact microbiome (the bacterial community that was dislodged after gentle washing) was determined with 16S rRNA amplicon sequencing. After sequence processing and quality control, the insect data set consisted of nine bumblebee samples (from three different batches) and seven predatory mite samples (from two different batches) for a total of 240,971 reads. On average, a bumblebee sample contained 20,371 (±11,844) reads, and a predatory mite sample contained 8,233 (±9,574) reads.

In total, we found 29 different ASVs on all samples of bumblebees, with an average of seven ASVs per sample. The genera *Snodgrasella* and *Gilliamella* were the most abundant in

**TABLE 1** All factors tested in two PERMANOVA models[a]

| Factor | PERMANOVA R² | PERMANOVA P-value | PERMDISP2 minimal P-value |
|---|---|---|---|
| Strawberries | | | |
| Department | 0.156 | 0.0 | 0.48 |
| Wk | 0.088 | 0.022 | 0.45 |
| | | | |
| Tomatoes | | | |
| Greenhouse | 0.088 | 0.001 | 0.01 |
| Department | 0.190 | 0.001 | 0.22 |
| Wk | 0.042 | 0.003 | 0.05 |

[a]Factors contributing to the phyllosphere compositional variation of strawberry and tomato samples. Strawberries samples were only collected from one greenhouse. Thus, the factor "Greenhouse" was not included in the model. The PERMANOVA P-value indicates that all factors were found to be significant ($P < 0.05$), and the R² value indicates the amount of variation explained by the respective factor. Permutational analyses for the homogeneity of multivariate dispersions (PERMDISP2) was done to test for the equality of variance between groups. The smallest observed P-value between groups is given.

the three batches (Fig. 4B). At the ASV level, three taxa occurred in all three batches of bumblebees, classified as *Snodgrasella* sp., *Gilliamella* sp., and *Lactobacillus bombicola* (Fig. 4B-2). For the predatory mites, the most abundant genera were different in the two batches (Fig. 4A), still 48 taxa were shared at the ASV level among the two batches (Fig. 4A-2). For the predatory mite samples, we found 434 ASVs in total, with an average of 100 ASVs per sample. One of the two batches of predatory mites contained a higher number of different ASVs (373 ASVs) compared to the other batch (109 ASVs). This could be explained by a deeper sequencing depth in that batch (on average, 13,167 reads per sample in one batch and 1,655 reads in the other batch). The inverse Simpson diversity index was not significantly higher in the first batch (on average, 7.86 and 3.27, $P = 0.127$).

**Overlap between the contact microbiome of bumblebees and mites and the phyllosphere microbiome.** We analyzed which taxa, at ASV level, were found in bumblebees and predatory mite samples, as well as in plant samples, as these taxa could have been present on the bumblebees and predatory mites that were present in the

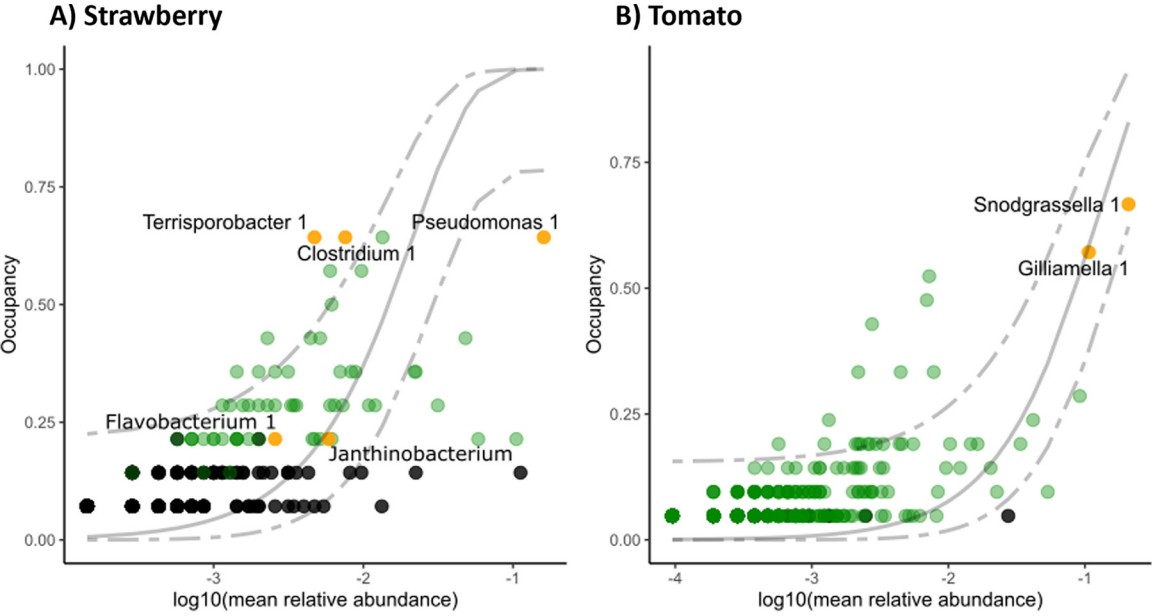

**FIG 3** Occupancy-abundance curves for the strawberry (A) and tomato samples (B). For each ASV, occupancy (proportion of plant samples in which an ASV was present) was plotted against the $log_{10}$ transformation of its mean relative abundance in these plant samples. ASVs that were identified as core taxa based on the 'elbow' method are highlighted in yellow, and their annotations are added (30). ASVs identifed as core taxa by the less stringent 'lastcall' method are highlighted in green. Other ASVs are shown in black. A neutral model was fitted on the occupancy-abundance curves (solid gray line shows the model fit, and dashed lines show the 95% confidence interval).

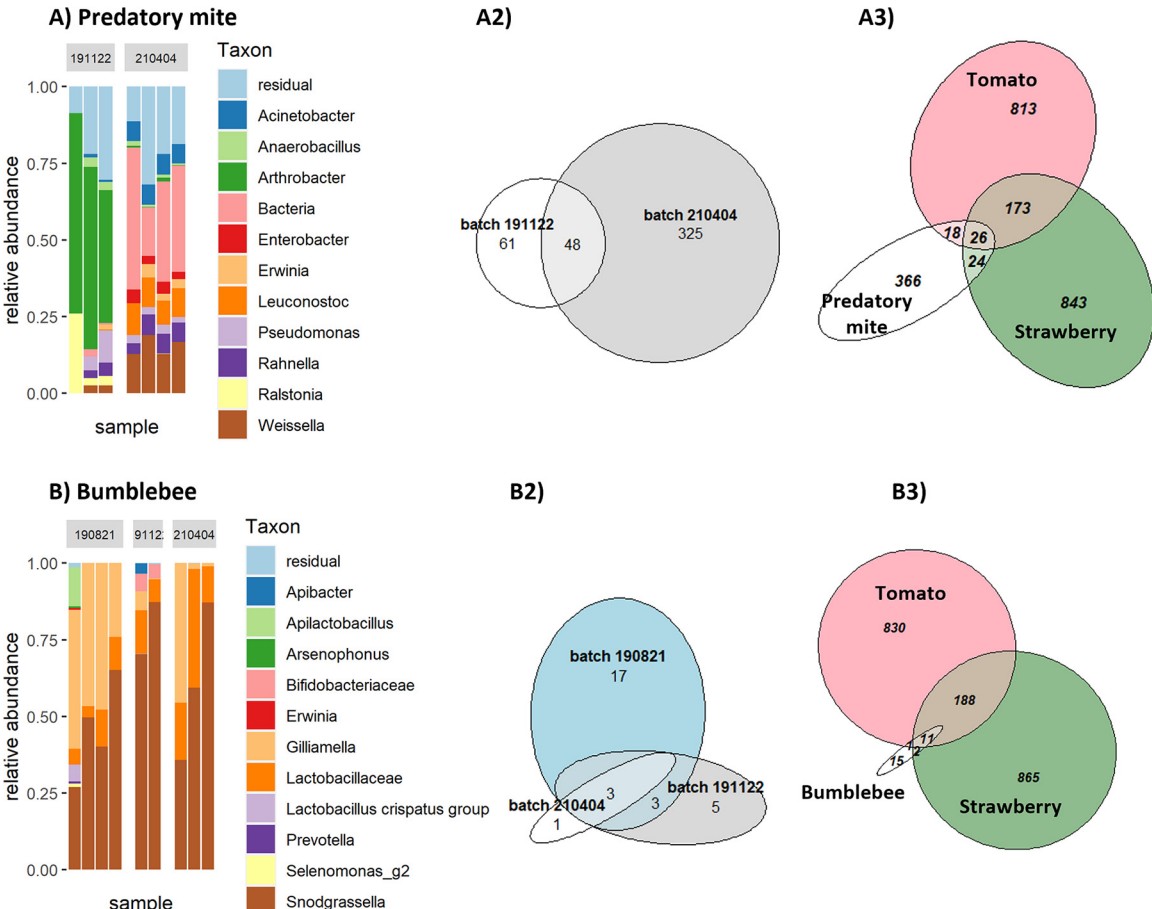

**FIG 4** Contact microbiome of different batches of predatory mites (A) and bumblebees (B). (1) Barplots showing the 11 most abundant genera for predatory mite (A) and bumblebee samples (B). Samples are grouped per batch, which were ordered at different times from the same company. (2) Venn diagrams showing the number of ASVs overlapping between different batches of predatory mites (A2) and bumblebees (B2). (3) Venn diagrams showing the number of ASVs overlapping between phyllosphere samples from tomato and strawberry, with predatory mites (A3) and bumblebees (B3).

greenhouses and thus could have introduced and dispersed these bacteria. However, since we did not have access to plants that did not come into contact with arthropods, we cannot exclude the possibility that these ASVs would not have been there without the arthropods. The bumblebee-associated ASVs with the highest occurrences in plants belonged to the genera *Snodgrasella* and *Gilliamella*. The ASV sequences that occurred on the bumblebees were identical to the ASVs found on the plants. The predatory mite-associated taxa that were most frequently found in plant samples belonged to the genera *Sphingomonas*, *Staphylococcus*, *Methylobacterium,* and *Pseudomonas*; however, this shared *Pseudomonas* ASV was not the same as the core ASV found in strawberry samples. The relative abundances of the arthropod-associated taxa in the plant samples are visualized in Fig. 5.

The impact of bumblebees and predatory mites on the phyllosphere microbiome was quantified using two parameters. First, the dispersal index was calculated to quantify the percentage of plants that had been impacted by arthropods. The dispersal index was defined as the percentage of plant samples that contained at least two arthropod-associated taxa (at the ASV level). Regarding bumblebees, 80% (57 out of 71) of the plant samples contained at least two bumblebee-associated taxa: 81% (17/21) for the strawberry samples and 80% (40/50) for the tomato samples. Predatory mites had a similar distribution range: 85% (60/71) of all plant samples contained predatory mite-associated taxa, with 90% (19/21) for the strawberry samples and 82% (41/50) for the tomato samples. Combined, only four phyllosphere samples, two strawberry

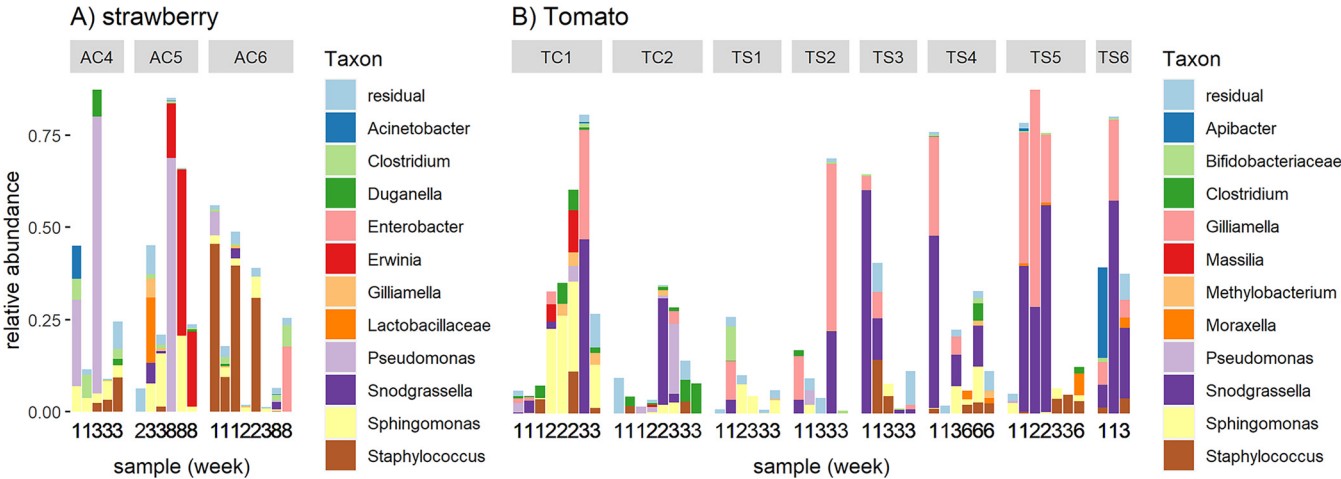

**FIG 5** Relative abundances of invertebrate-associated ASVs in strawberry (A) and tomato (B) samples. The ASVs are aggregated at the genus level, and only the 11 most abundant genera are shown, with the remaining taxa being grouped in 'residual'. Similarly to Fig. 1C and D, the plant samples are divided by department, and the sampling week is indicated by a number underneath each bar. Two strawberry and two tomato samples did not contain at least two reads of an invertebrate-associated taxon and are not shown in the figure.

samples and two tomato samples, did not contain at least two arthropod-associated taxa. These samples are not shown in Fig. 5. Second, we calculated the transfer index for samples that contained arthropod taxa as the share of reads (%) in a plant sample that were identical to the ASVs found in bumblebee and/or predatory mite samples. We observed that, on average, 10.9% of the reads from the strawberry samples and 26.0% of the reads from the tomato samples that contained at least two bumblebee-associated reads were bumblebee-associated reads. For predatory mites, on average, 32.7% of the reads from the strawberry samples and 28.5% of the reads from the tomato samples were predatory mite-associated ASVs in samples that contained these taxa. Additionally, there was a high variation in the relative abundance, as well as in the taxonomy, of arthropod-associated taxa between phyllosphere samples (Fig. 5).

## DISCUSSION

In this study, we confirmed that the phyllosphere microbial communities in greenhouses were low in density and diversity. Our 16S rRNA amplicon sequencing of phyllosphere samples resulted in low bacterial read numbers. The majority of reads were discarded, as they were annotated as mitochondrial and chloroplast DNA. To avoid the amplification of such nonbacterial DNA, PNA clamps were used during PCR in subsequent sequencing runs (see Materials and Methods, [9]). The samples that were amplified and sequenced without the use of PNA clamps consisted, on average, of 95.5% and 92.5% nonbacterial reads in tomato and strawberry samples, respectively. These ratios were high compared to the results of most other studies using the same techniques, which reported percentages between 0 and 90% (2, 8, 9). As the host versus bacterial DNA ratio can be used as an indication for absolute abundance of bacteria on the phyllosphere (8), this high ratio indicates that the bacterial biomass in the samples in this study was relatively low or that the leaves were fragile and the plant cells were easily ruptured. The addition of both plastid and mitochondrial blocking PNA clamps to the PCR reduced the proportion of nonbacterial reads to 10.3% in the strawberry samples and 29.1% in the tomato samples. However, absolute bacterial read numbers (on average, 4,394 for the strawberry samples and 1,282 for the tomato samples) remained low in comparison to other studies using similar methods (typically $10^4$ to $10^5$ bacterial reads per sample [2, 10, 11]). From this, we can conclude that the addition of PNA clamps was successful in reducing the amplification and, therefore, the

sequencing of host-plant DNA, but it did not resolve the problem of low bacterial read numbers.

Culture-dependent techniques confirmed that bacterial densities were low. On average, the samples contained $8 \cdot 10^2$ CFU/g, which was similar to bacterial densities previously described for laboratory-grown lettuce (4). In comparison, the study by Williams and Marco found 10- to 100-fold more CFU on field-grown lettuce compared to laboratory-grown plants.

In addition to a low abundance, the bacterial diversity was low in the phyllosphere samples. Often, only a few taxa dominated a sample, and these taxa were different across departments and over time. For example, the genus *Acinetobacter* dominated the tomato samples in one department (TC2) over the course of 3 weeks, but it did not disperse to other departments. Dong and colleagues (2019) had also observed that this genus made up 97% of all reads in the phyllosphere microbiome of their greenhouse-grown tomato plants. Such observations, where a single ASV dominates the tomato or strawberry phyllosphere of only a few samples, are generally less common in less controlled environments (3, 10).

To further understand which taxa played an important role in shaping the phyllosphere community, we identified the core taxa (with a method based on Shade and Stopnisek [8]). In the strawberry samples, five core ASVs were identified, which belonged to the genera *Clostridium*, *Pseudomonas*, *Terrisporobacter*, *Flavobacterium*, and *Janthinobacterium*. *Clostridium* and *Terrisporobacter* are spore-formers that persist in various environments, including soil and rhizosphere (11, 12). These two ASVs were widespread and were detected in all three greenhouse groups (Fig. 1), including the tomato samples from the second greenhouse, where no strawberries were grown. The other three core ASVs, *Pseudomonas*, *Janthinobacterium*, and *Flavobacterium*, are also commonly found in the phyllosphere (13–16). Moreover, these genera have been found in rain and were enriched in the phyllosphere of plants exposed to this rain (10).

The core taxa in tomato samples belonged to the genera *Snodgrasella* and *Gilliamella*. This is remarkable, as these genera are usually associated with the bumblebee microbiome (17, 18). Indeed, sequencing the contact microbiome of commercially-reared bumblebees showed that the exact same ASVs were found in these bumblebees. The two tomato core ASVs also occurred in all three greenhouse groups, and they usually co-occurred in a sample. These findings indicate that bumblebees introduce and disperse these bacteria in greenhouses. Next to these two core taxa, we found that the ASVs annotated as *Bifidobacteriaceae*, *Clostridium*, and *Lactobacillus* occurred in more tomato samples than expected based on their abundance. The same ASVs also occurred on bumblebees, suggesting that these were also dispersed by bumblebees.

Compared to the tomato samples, the impact of bumblebees on the microbiome of the strawberry plants was less profound. While the dispersal index was similar (81%) for strawberry crops, a smaller share of the reads on the strawberry plants were annotated as bumblebee-associated ASVs (10.9%). Furthermore, the genera *Snodgrassella* and *Gilliamella* did not occur in high abundances in the strawberry samples and were not considered core taxa in strawberries. The strawberry microbiome was thus less receptive to bumblebee-associated reads. Furthermore, we also found a stronger relationship between the occupancy and abundance of taxa in the strawberry samples compared to the tomato samples, which suggests a more developed community in the strawberry samples compared to the tomato samples and could explain the lower impact of the bumblebee-associated taxa on the strawberry microbiome. Differences in cultivation practices of strawberry crops versus tomato crops could explain why the strawberry microbiome was more robust. Strawberry plants were replicated vegetatively (versus from seed). They were grown in substrate based on peat (versus in rockwool), and their leaves were physically closer to each other and to the substrate. These factors could have enhanced the transfer and selection of plant-adapted bacteria, leading to a more developed and robust microbiome which was less receptive to bacteria introduced by bumblebees.

Compared to bumblebees, little is known about the microbiome of predatory mites. To our knowledge, this is the first report on 16S rRNA amplicon sequencing of *Phytoseiulus persimilis*. An ASV from the genus *Arthrobacter* dominated the microbiome in one batch, while an unknown bacterium dominated the second. Blasting this unknown ASV sequence on *EzBioCloud* (ezbiocloud.net) did not result in any further hits. This illustrates that the microbiome of predatory mites is still largely unexplored.

Moreover, there was a large difference in community composition between the two batches of predatory mites. This could be explained by the rearing process of these arthropods, which are fed with plant material and/or prey species living on these plants (19). These feed sources are themselves colonized by different microbial communities and likely result different microbiomes between batches. As a different batch of predatory mites had been released in the greenhouses and plants could not be sampled before they had come into contact with predatory mites, strong conclusions on which microbes were introduced and dispersed by predatory mites could not be made. Nevertheless, a high number of ASVs were shared between the phyllosphere and the predatory mites, suggesting that there is potential for these arthropods to introduce and disperse bacteria in greenhouses. We found that 85% of all plant samples contained at least two predatory mite-associated ASVs, and these ASVs made up approximately one-third of the reads in the strawberry samples (32.7%) and the tomato samples (28.5%). Looking at the taxonomy of these ASVs, we saw that these are genera that are commonly found in the phyllosphere (20), including *Sphingomonas*, *Methylobacterium*, and *Pseudomonas*. The high number of overlapping ASVs and their taxonomy suggest that predatory mites could harbor phyllosphere-adapted bacteria and play a role in shaping the phyllosphere microbiome. However, due to the limitations of this study, primarily its lack of a control group, it was not possible to conclude which taxa had been introduced and dispersed by predatory mites.

Another atypical member of the phyllosphere microbiome was the genus *Portiera*, which occurred in two tomato departments at various time points. This genus is known as a primary endosymbiont of whitefly (21), a pest in greenhouses. The presence of these reads in the phyllosphere samples indicates that whiteflies (including eggs or instars) were present on the leaves. Indeed, the two departments in which *Portiera* reads were abundant suffered from a high number of whiteflies, according to sticky cards, a common method used to monitor pest pressure in greenhouses (22). This shows that the presence of *Portiera* in the phyllosphere microbiome could be further investigated as a marker for whitefly populations in greenhouses, possibly allowing for earlier detection and/or a reduction of labor compared to the current method of using sticky cards.

Finally, this study also revealed a possible risk in the dispersal of bacteria by arthropods. Both bumblebees and predatory mites contained a different ASV assigned to the genus *Erwinia*, and these two ASVs also occurred on plants. Some species in the *Erwinia* genus are phytopathogens, while other species are known biocontrol agents. Further studies are needed to assess the risk of introducing phytopathogens by arthropods used in greenhouses.

In conclusion, this study revealed that the phyllosphere microbiome of greenhouse crops is generally low in abundance and diversity. Furthermore, we found many identical ASVs on crops and on beneficial arthropods that are commonly used in greenhouses. Despite the limitations of this study, mainly the lack of a control group in which no arthropods were released, we can conclude that bumblebees introduced and dispersed bacteria in greenhouses, specifically the genera *Snodgrasella* and *Gilliamella*. Moreover, we found that the contact microbiome of predatory mites consisted of a wide diversity of phyllosphere-associated ASVs, many of which were identical to ASVs found on plants. Further experimental studies could reveal to what extent the phyllosphere microbiome is impacted by the release of beneficial arthropods, and such knowledge could be applied to diversify the greenhouse microbiome and thereby improve plant health.

## MATERIALS AND METHODS

**Sampling.** Samples were collected from tomato (*Lycopersicum solanum*) and strawberry (*Fragaria x ananassa*) plants in two research greenhouses in Flanders: the Research Station for Vegetable Production (Proefstation voor de Groenteelt, in Sint-Katelijne-Waver; greenhouse S) and Research Centre Hoogstraten (Proefcentrum Hoogstraten, in Hoogstraten; greenhouse C). All crops were treated following the Integrated Pest Management (IPM) principles, using standard chemical and biological solutions to prevent pests and diseases. Crops in the same department received the same treatments and were planted at the same time. Departments were separated by glass walls. Different commercial cultivars were grown in the different departments of the two greenhouses: Foundation, HTL1606899, HTL1606377, Rebelski, Kanavaro, Ezanzo, and Merlice for tomatoes and Elsanta, Clery, and Malling Centenery for strawberries. Plants were sampled over 8 weeks in March and April 2019. At different time points, samples were taken from the same rows and the same departments but not necessarily from the same plant.

For each sample, approximately 4 g of leaves were collected in a 50 mL tube (Greiner Bio-One), using gloves and scissors sterilized with 70% ethanol. The samples were kept on ice for transportation to the lab. In the lab (at most 4 h after sampling), 5 mL of a 1:50 diluted leaf wash buffer (1 M tris-Hcl, 500 mM EDTA, 1.2% Triton, adjusted to pH 8 [23]) was added to the tubes with leaves. The tubes with leaves and leaf wash buffer were vortexed for 5 min at maximum speed with the Vortex Genie 2 (MoBio), to suspend the bacteria. The tubes were then centrifuged at 1,000 g for several seconds to spin down most of the buffer from the leaves, which were then removed from the tubes.

For samples of which the colony forming unit (CFU) count was determined, the leaf wash was plated out on nonselective Reasoners 2A (R2A) agar medium supplemented with cycloheximide (final concentration 0.1 g/L) to suppress fungal growth. Plates were incubated at room temperature (approximately 22°C) for 2 days, after which CFUs were counted.

The remaining leaf wash buffer was then centrifuged at 12,000 g for 2 min in aliquots of 2 mL to harvest the bacterial cells. The pellets were resuspended in 750 $\mu$L of Power Bead Solution (QIAmp Powerfecal DNA kit Qiagen), and stored at −20°C until further processing.

Bumblebees (*Bombus terrestris*) and predatory mites (*Phytoseiulus persimilis*) were ordered directly from a breeding facility at Biobest NV (www.biobestgroup.com, Ilse Velden 18, 2260 Westerlo, Belgium). A total of 11 bumblebee samples were taken from three different batches (ordered on August 28, 2019, November 22, 2019 and April 4, 2021), and eight predatory mite samples from two batches (ordered on November 22, 2019, and April 4, 2021). Samples of the contact microbiome of the bumblebees were taken by letting the bees exit their hive and walk into sterile 50 mL tubes (Greiner Bio-One). Two to four bumblebees were pooled in one tube per sample. Samples of the contact microbiome of predatory mites were taken by transferring approximately 4 g of mites, including the substrate (sawdust) in which they were transported, into 50 mL tubes (Greiner Bio-One). Next, the sampled bumblebees and predatory mites were washed with leaf wash buffer, similar to the phyllosphere samples. Instead of removing the leaves, 4 mL of the leaf wash buffer was pipetted out of the tubes, after allowing larger particles to settle on the bottom. We defined the bacterial community that was dislodged from the arthropods during gentle washing as the contact microbiome and assume that this would be comparable to the bacterial community to which the phyllosphere would be exposed.

**DNA sequencing.** The DNA of the phyllosphere and invertebrate samples was extracted using the PowerFecal DNA isolation kit (Qiagen), according to the manufacturer's instructions, except for the final elution step, which was instead performed with 60 $\mu$L elution buffer to increase the final DNA concentration of the extracts. Two blanks per extraction kit were included: one at the at the start (leaf wash buffer washing step) and one at the end of each kit.

Next, the extracted DNA was amplified using barcoded primers (IDT), as described by Kozich et al. The V4 region of the 16S rRNA gene was amplified in 30 cycles in a 20 $\mu$L reaction with Phusion High-Fidelity DNA polymerase (Thermo Fisher Scientific). Some phyllosphere extracts were amplified and sequenced two or three times to optimize the number of bacterial reads per sample. Later, only the replicate with the highest number of bacterial reads would be retained. In a second amplification and sequencing run, peptide nucleic acid (PNA) clamps, designed to specifically bind and block the amplification of plastid DNA, were added (pPNA, 5′-GGCTCA ACCCTGGACAG-3′) (24). As nonbacterial read numbers remained high, mitochondrial PNA clamps were also added in a subsequent run (run 4) (mPNA, 5′-GGCAAGTGTTCTTCGGA-3′). Invertebrate samples were amplified and sequenced without the addition of PNA clamps in two separate runs (run 3 and run 5). Each PCR included two PCR blanks, which were also sequenced. Later, all sequencing data were merged, and only the samples with the highest numbers of reads were kept (see data processing). Cycling conditions during PCR were: initial denaturation at 95°C for 2 min, 30 cycles at 95°C for 20 s, 75°C for 10 s (only with the addition of PNA clamps, clamping temperature), 55°C for 20 s, 72°C for 1 min, and a final extension at 72°C for 10 min. Two PCR blanks were included. Next, the amplicons were purified using Ampure XP (Beckman Coulter). and the DNA concentration of the purified samples was quantified using a Qubit 3.0 fluorometer (Life Technologies). These DNA concentrations were used to pool samples and blanks in equimolar concentrations, resulting in a library. The amplicon library was further purified by loading it on a 0.8% (mass/vol) agarose gel and extracting bands of approximately 380 bp with the Nucleospin Gel and PCR Clean-up kit (Macherey-Nagel). The final library was diluted to 2 nM and sequenced on the Illumina MiSeq platform using 2 × 250 cycles at the Center of Medical Genetics Antwerp (University of Antwerp, Belgium). The sequencing data of this study were made available under study accession number PRJEB43218 in the European Nucleotide Archive.

**Data processing.** The raw sequencing data were processed with the package DADA2 (25) in R. In brief, reads with more than two expected errors were removed. Forward and reverse reads were denoised per sample using the DADA2 algorithm, and reads were merged. Chimeras were removed using the removeBimeraDenovo function, and a table with ASVs was constructed. The ASVs were

classified using the EZBiocloud reference 14S rRNA database (26). Nonbacterial reads (i.e., plastid and mitochondrial DNA) were removed from the data set. Specific ASVs were identified as contaminants based on their presence in the blanks in combination with their low presence in the phyllosphere samples (less than 10 reads) and their likelihood to be common contaminants (Table 2). Prevotella 1 occurred in high abundances in airway samples sequenced on the same run and was also considered a contaminant. ASVs in blanks from run 2 and run 3 were present in higher numbers in the samples than in the blanks and were not considered contaminants. Blanks in run 4 and run 5 did not contain any bacterial reads.

Next, the five sequencing runs were merged. As described above, some samples were sequenced two or three times, with and without the addition of PNA clamps. Only the replicate with the highest read number for each sample was kept after merging. An overview of the important parameters of each sequencing run is given in Table 3. A permutational multivariate analysis of variance was performed to evaluate the effect of the sequencing run on the community composition of the samples. For the strawberry samples, there was no significant effect ($R^2 = 0.13$, $P = 0.082$). For the tomato samples, week and department were confounding factors, as the samples in different runs were from different weeks and departments. Therefore, we looked at the effect of sequencing run only for the double- or triple-sequenced samples. For these samples, there was no significant effect of sequencing run ($R^2 = 0.02302$, $P = 0.101$).

Finally, taxon names were assigned to the ASVs. Samples containing fewer than 100 reads were discarded. Further data analysis was done in the R environment, using the tidyverse set of packages (27) and an in-house built package, tidyamplicons (https://github.com/SWittouck/tidyamplicons). Alpha diversity indices and sample library sizes (numbers of reads) between sample groups were compared using Student's $t$-tests (in cases in which the data were normally distributed) or the Wilcoxon test (in cases in which the data were not normally distributed). Normality was tested using the Shapiro test. Graphs were generated using the ggplot2 package (28) and the Eulerr package (29).

Core taxa were defined based on the occupancy, consistency, and contribution of the taxa to the beta diversity in the entire community, as suggested by Shade and Stopnisek (30). This analysis was done separately for the tomato and strawberry samples, and each data subset was rarefied (using the vegan package in R [31]) to 500 reads to avoid bias from samples which were sampled to a much greater depth. This resulted in reducing 50 tomato samples containing 1,030 different ASVs to 21 samples containing 514 ASVs. The strawberry samples were reduced from 21 to 14 samples and from 1,066 to 452 ASVs. First, the ASVs were ranked according to their time-specific occupancy and consistency in replicates, which were samples within the same department. Next, Bray-Curtis dissimilarities were calculated as a measure of beta diversity between samples from the same plant species. Prospective core sets were formed by starting from the top-ranked taxa and consecutively adding the next-ranked taxon. The Bray-Curtis dissimilarity for each prospective core set was then divided by the Bray-Curtis dissimilarity for the complete data set to calculate the contribution of the prospective core set to the overall beta diversity. Next, the 'elbow' method, based on the maximized first-order difference between the two parts of the cumulative contribution curve, was used to identify which taxa contributed most to the beta diversity, and these were identified as core taxa (30). A neutral model was fitted on the occupancy-abundance curves, using R code based on Burns et al. (32). The neutral model assumes that community assembly is the result of stochastic processes and is used to discriminate taxa that are deterministically selected (either by selection or dispersal limitation).

To quantify the overlap between the arthropod and phyllosphere microbiomes, two indices were calculated. First, the dispersal index (%) was defined as the share of phyllosphere samples that contained at least two reads of ASVs that also occurred on the arthropods. Second, the transfer index (%) was defined as the share of reads in the phyllosphere samples that were annotated as invertebrate-associated ASVs. Phyllosphere samples that did not contain any invertebrate taxa were excluded from this calculation. An ASV was considered present in an invertebrate sample if it had two or more reads.

**Data availability.** The sequencing data of this study were made available under study accession number PRJEB43218 in the European Nucleotide Archive.

**TABLE 2** ASVs identified in blanks of each run[a]

| Taxon ID | Run | Max no. of reads in samples | Max no. of reads in blanks | Removed as contaminant? |
|---|---|---|---|---|
| *Staphylococcus* 1 | Run 1 | 10 | 2 | Yes |
| Prevotella 1 | Run 1 | 406 | 16 | Yes |
| Fusobacterium 1 | Run 1 | 10 | 4 | Yes |
| Fusobacterium 2 | Run 1 | 6 | 13 | Yes |
| Caulobacter 1 | Run 1 | 6 | 9 | Yes |
| Lactobacillus 1 | Run 2 | 79 | 74 | |
| Bifidobacterium | Run 2 | 81 | 18 | |
| Lactobacillus 2 | Run 2 | 54 | 48 | |
| Snodgrasella | Run 3 | 22554 | 3 | |
| Ralstonia | Run 3 | 82 | 59 | |

[a]For each ASV, the maximum numbers of reads in samples and in blanks are given. Contaminants were identified based on their presence in the blanks in combination with their low presence in the phyllosphere samples (fewer than 10 reads).

**TABLE 3** Overview of important parameters for each sequencing run

| Run | Before removing nonbacterial reads | | | After removing nonbacterial reads | | | | | | | Sample types[d] | PCR conditions |
|---|---|---|---|---|---|---|---|---|---|---|---|---|
| | No. reads | No. reads pPNA[a] | No. reads mPNA[b] | No. reads | No. reads pPNA[a] | No. reads mPNA[b] | No. samples | No. taxa[c] | Avg library size | % nonbacterial | | |
| Run 1 | 5E+06 | 4E+06 | 5E+05 | 9E+04 | 0 | 0 | 85 | 1,289 | 1,033 | 96.6 | P | No clamps |
| Run 2 | 5E+05 | 6E+03 | 3E+05 | 6E+04 | 2 | 0 | 94 | 761 | 1,089 | 81.0 | P | pPNA clamps |
| Run 3 | 1E+05 | 1E+03 | 3E+02 | 1E+05 | 0 | 0 | 12 | 142 | 9,487 | 3.29 | I | No clamps |
| Run 4 | 8E+04 | 1E+03 | 2E+03 | 6E+04 | 0 | 0 | 58 | 761 | 1,114 | 39.6 | P | pPNA and mPNA clamps |
| Run 5 | 1E+05 | 0E+00 | 0E+00 | 1E+04 | 0 | 0 | 7 | 377 | 20,569 | 0.0 | I | No clamps |

[a]The number of reads compatible with plastid peptide nucleic acid (pPNA) clamps.
[b]Mitochondrial peptide nucleic acid (mPNA) clamps.
[c]Number of different amplicon sequence variants (ASVs)
[d]Sample types included in the run (P = phyllosphere samples, I = invertebrate samples).

## ACKNOWLEDGMENTS

We thank the Centre for Medical Genetics Antwerp for the use of their Illumina MiSeq sequencing system. We are also grateful to collaborators at the two research centers in Hoogstraten and Sint-Katelijne Waver Hoogstraten for their help and guidance during sampling.

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
