## [Reviewer comments · Microbiology Spectrum]

Microbiology Spectrum

the greenhouse phyllosphere microbiome and associations with introduced bumblebees and predatory mites

Marie Legein, Wenke Smets, Karen Wuyts, Lien Bosmans, Roeland Samson, and Sarah Lebeer

Corresponding Author(s): Sarah Lebeer, University of Antwerp

Review Timeline:

Submission Date:	May 16, 2022
Editorial Decision:	May 22, 2022
Revision Received:	June 13, 2022
Accepted:	June 17, 2022

Editor: Kristen DeAngelis

Reviewer(s): The reviewers have opted to remain anonymous.

Transaction Report:

DOI: <https://doi.org/10.1128/spectrum.01755-22>

May 22, 2022

Prof. Sarah Lebeer
University of Antwerp
Bioscience Engineering
Groenenborgerlaan 171
Antwerpen 2020
Belgium

Re: Spectrum01755-22 (The impact of introduced bumblebees and predatory mites on the phyllosphere microbiome of greenhouse crops)

Dear Prof. Sarah Lebeer:

Thank you for submitting your manuscript to Microbiology Spectrum. I think this could be a good contribution to our journal, but I agree with the editor from mSystems that the hypothesis is not a good match to the experimental design, results and discussion. I would like for you to revise your manuscript and make a point-by-point rebuttal of the previous editors comments, including addressing the caveats to the study in the discussion.

Link Not Available

Sincerely,

Kristen DeAngelis

Journals Department
Reviewer comments:

Staff Comments:

Preparing Revision Guidelines

To submit your modified manuscript, log onto the eJP submission site at <https://spectrum.msubmit.net/cgi-bin/main.plex>. Go to

Author Tasks and click the appropriate manuscript title to begin the revision process. The information that you entered when you first submitted the paper will be displayed. Please update the information as necessary. Here are a few examples of required updates that authors must address:

Please return the manuscript within 60 days; if you cannot complete the modification within this time period, please contact me. If you do not wish to modify the manuscript and prefer to submit it to another journal, please notify me of your decision immediately so that the manuscript may be formally withdrawn from consideration by Microbiology Spectrum.

Response to Reviewers

Decision letter Spectrum01755-22 on 23/05/2022

Thank you for submitting your manuscript to Microbiology Spectrum. I think this could be a good contribution to our journal, but I agree with the editor from mSystems that the hypothesis is not a good match to the experimental design, results and discussion. I would like for you to revise your manuscript and make a point-by-point rebuttal of the previous editors comments, including addressing the caveats to the study in the discussion.

Authors: Dear editor, Thank you for giving us the chance to adapt our manuscript. Please find below a point-by-point rebuttal of the previous editors comments. We have also adapted the manuscript and addressed the caveats of the study in the discussion. Important changes have been highlighted in the Marked-Up manuscript.

Decision letter mSystems00384-22 on 27/04/2022

The paper was reviewed internally by 3 experts who felt that the experimental design fell short of being able to test the primary hypothesis as presented.

This is very good idea which suffers from a poor experimental planning. The main problem is the lack of untreated plants. Likewise, it is unclear whether the titre of insects was the same for the different crops and/or in the various departments (or was quantified at all) as this will likely impact on what is the "input" microbiota. As the plant genotype is a determinant of the microbiota, you could have quantified its impact on the mite/insect effect. The results seem slightly descriptive, but have the potential to systematically dissect how these insect might alter the phyllosphere microbiome. However, without sampling plants before adding the insects, you cannot quite tell if part of the plant microbiome might originate from insect (which was assayed before adding these to the plants).

Point-by-point rebuttal

The paper was reviewed internally by 3 experts who felt that the experimental design fell short of being able to test the primary hypothesis as presented.

Authors: We agree with the reviewer and want to clarify that the study started as an observational study and was not designed to test this hypothesis. We have adapted the abstract, which now describes the study better as an observational and explorative study and removed that a specific hypothesis was tested.

This is very good idea which suffers from a poor experimental planning. The main problem is the lack of untreated plants.

Authors: Indeed, we agree that this study lacks a control group. Since this was an observational study done in greenhouses in which different cultivars were tested and where the yield was destined for the commercial market, there was no option to

change the management of these greenhouses. Nevertheless, even without this control, we could make sufficient relevant conclusions based on other comparisons, mostly on the impact of bumblebees, which is discussed on Ln 239-248. For the impact of predatory mites, we agree that some more nuance was needed, as discussed now on Ln 275 -288. Finally, we added a concluding alinea (Ln 304-314) which highlights the caveats of the study.

Likewise, it is unclear whether the titre of insects was the same for the different crops and/or in the various departments (or was quantified at all) as this will likely impact on what is the "input" microbiota.

Authors: This was indeed not quantified, because of practical considerations, it was not possible to control for and measure this. This limitation of the study design is now made more clear in the introduction (Ln 72-74) and the results section (Ln 139 – 146 and Ln 168-170). Nevertheless, even with taken this bias into account, the conclusion that specific taxa were introduced in the greenhouses by bumblebees is still valid

As the plant genotype is a determinant of the microbiota, you could have quantified its impact on the mite/insect effect.

Authors: The reviewer also made a valid point that plant genotype is important. However, all plants used here were commercial cultivars, postulated to have a similar genotype (strawberries vs tomatoes), so that this impact is postulated to be minor. Moreover, to quantify the differences between strawberry and tomato, a more controlled introduction of beneficial arthropods would be needed.

The results seem slightly descriptive, but have the potential to systematically dissect how these insect might alter the phyllosphere microbiome. However, without sampling plants before adding the insects, you cannot quite tell if part of the plant microbiome might originate from insect (which was assayed before adding these to the plants).

Authors: We agree with the reviewer that due to the lack of a control group, we need to remain descriptive. Yet, some interesting novel observations were obtained, such as the transfer of *Gilliamella* and *Snodgrassella* from bumblebees to tomato plants, the presence of *Portiera* in whitefly infested departments, and the presence of typical phyllosphere-associated bacteria on predatory mites. These observations are worth communicating about, because they inspire novel research. Novel experiments should best be performed in more controlled/experiment environments, outside commercial greenhouses as we did, because the need for more controls.

June 17, 2022

Prof. Sarah Lebeer
University of Antwerp
Bioscience Engineering
Groenenborgerlaan 171
Antwerpen 2020
Belgium

Re: Spectrum01755-22R1 (the greenhouse phyllosphere microbiome and associations with introduced bumblebees and predatory mites)

Dear Prof. Sarah Lebeer:

Your manuscript has been accepted, and I am forwarding it to the ASM Journals Department for publication. You will be notified when your proofs are ready to be viewed.

Sincerely,

Kristen DeAngelis
Editor, Microbiology Spectrum
